# Nasal Tumor Vaccination Protects against Lung Tumor Development by Induction of Resident Effector and Memory Anti-Tumor Immune Responses

**DOI:** 10.3390/pharmaceutics15020445

**Published:** 2023-01-29

**Authors:** Michael Donkor, Jamie Choe, Danielle Marie Reid, Byron Quinn, Mark Pulse, Amalendu Ranjan, Pankaj Chaudhary, Harlan P. Jones

**Affiliations:** 1Department of Microbiology, Immunology and Genetics, University of North Texas Health Science Center, Fort Worth, TX 76107, USA; 2Department of Biology, Langston University, Langston, OK 73050, USA; 3Department of Pharmaceutical Sciences, University of North Texas Health Science Center, Fort Worth, TX 76107, USA

**Keywords:** breast cancer, lung metastasis, vaccination, immune response, nanoparticles

## Abstract

Lung metastasis is a leading cause of cancer-related deaths. Here, we show that intranasal delivery of our engineered CpG-coated tumor antigen (Tag)-encapsulated nanoparticles (NPs)—nasal nano-vaccine—significantly reduced lung colonization by intravenous challenge of an extra-pulmonary tumor. Protection against tumor-cell lung colonization was linked to the induction of localized mucosal-associated effector and resident memory T cells as well as increased bronchiolar alveolar lavage-fluid IgA and serum IgG antibody responses. The nasal nano-vaccine-induced T-cell-mediated antitumor mucosal immune response was shown to increase tumor-specific production of IFN-γ and granzyme B by lung-derived CD8^+^ T cells. These findings demonstrate that our engineered nasal nano-vaccine has the potential to be used as a prophylactic approach prior to the seeding of tumors in the lungs, and thereby prevent overt lung metastases from existing extra pulmonary tumors.

## 1. Introduction

Most cancer deaths are attributed to metastasis to distal organs and not from the primary tumor mass itself. Cancer metastasis to the lung occurs explicitly in 20 to 54% of patients with the malignant disease [1], and represents one of the leading causes of morbidity and mortality among cancer patients with metastatic disease. Furthering our understanding of how distant primary tumors metastasize to the lung and other metastatic sites should serve as a basis for creating therapeutic interventions that will ultimately increase survivorship in cancer patients.

The metastatic cascade is a multi-step process. The first step involves the dislodgement of tumor cells from the primary tumor into circulation, where they exist as circulating tumor cells (CTCs) [2]. Most CTCs do not survive the sheer force of blood and cellular immune defenses capable of seeding into metastatic organs where they exist as disseminated tumor cells (DTCs) [3]. However, DTCs that reach their target organ will remain dormant and undetectable by diagnostic techniques until their microenvironment becomes permissive to support their outgrowth, making organ colonization the rate-limiting step of the metastatic process [4,5,6,7]. The formation of microenvironments that support tumor metastasis is a continuation of target organ reprogramming, often referred to as the ‘pre-metastatic niche’ that prepares the seeding and establishment of DTCs [8,9]. Pre-metastatic niche formation begins with secreted components from primary tumors known as tumor-derived secretory factors (TDSF), comprised of cytokines and extracellular vesicles carrying varied molecules [9,10]. TDSFs home into the metastatic organ and simultaneously act on the bone marrow to recruit bone marrow-derived cells (BMDCs). The interplay between TDSFs, BMDCs, and the stroma of the metastatic organ creates the pre-metastatic niche to support DTCs’ extravasation into the organ. In addition, DTCs, TDSFs, and BMDCs coordinate within the pre-metastatic niche to promote immune-dysregulating machinery that dampens the tumor immunosurveillance creating a convivial environment for future DTCs entrants. Such immune perturbations result in an expansion of immature immunosuppressive cell types, including neutrophils (known as polymorphonuclear myeloid-derived suppressor cells) and monocytes (known as mononuclear myeloid derived-suppressor cells) as well as regulatory B- and T cells, whose activities shift the immune landscape of metastatic organs towards a suppressive state that compromises the organs’ antitumor immune defenses [11,12]. This tumor-induced immune dysregulation is problematic for the lung because of the organ’s intrinsic necessity to support a tolerogenic environment to protect against unwarranted inflammation triggered by exposure to innocuous particles and allergens from the air. This natural “tolerogenicity” makes it susceptible to extra-pulmonary tumors to establish immunosuppressive niches in the lung to drive secondary lung metastasis.

Current conventional treatments for most primary tumors also elicit adverse off-target effects on immune populations. For example, chemotherapy and radiation are known to impair the function as well as reduce the populations of antigen-presenting cells (APCs) and T cell [13]. In addition, surgery also induces immune perturbations by several mechanisms leading to reduced tumor immune surveillance [14]. Thus, while conventional treatments can successfully eradicate primary tumors, they fail to prevent and/or reduce metastasis.

A viable strategy to enforce DTCs’ removal from the pre-metastatic niche and the eventual inhibition of secondary lung metastasis could be to pre-empt the immunosuppressive microenvironment induced by the tumor burden in the lung. Vaccination strategies have been established to directly induce antitumor immune responses by stimulating endogenous antitumor T cells capable of long-term memory. However, the clinical benefits of cancer vaccines have not yet been fully realized due to a focus on evoking local immune responses within the primary tumor itself. A limitation of this strategy is that tumors at the time of diagnosis are already established with an increased immunosuppressive environment and utilize mechanisms to prevent vaccine-elicited T cells from accumulating within the solid tumor and its microenvironment. In addition, antitumor T cells that defy the tumor suppressive barriers and accumulate in primary tumors are quickly rendered anergic in the context of the immunosuppressive environment and prolonged interaction with primary tumor cells [15,16]. Therefore, we postulate that deploying cancer vaccines targeting metastatic sites would protect against metastatic cancer development.

We have previously formulated a cancer vaccine based on nanotechnology by taking advantage of the well-documented pathogen-associated molecular patterns (PAMPS); CpGs to produce immune-activating tumor antigen-encapsulating nanoparticles (CpG-NP-Tag). Intraperitoneal administration of the nanoparticle (NP) formulation was found to slow the primary breast tumor growth in mice that were challenged subcutaneously with 4T1 breast tumor cells from which the tumor antigens (i.e., membrane fractionated proteins) were derived [17,18]. Furthermore, the reduction in tumor growth was associated with an increased accumulation of CD4^+^ and CD8^+^ T cells within the solid tumors. These findings demonstrated the efficacy of our NP-based immunostimulatory vaccine against primary tumors. Nonetheless, since most solid tumor deaths result from metastasis to vital organs, as previously mentioned, this study tested the utility of our NP-based immunostimulatory vaccine for targeted delivery across the respiratory tract. The expectation is that delivery across the respiratory tract using intranasal vaccination will induce robust mucosal antitumor immunity providing increased resistance against primary tumor-cell colonization in the lungs.

## 2. Materials and Methods

### 2.1. Materials and Reagents

Poly lactide glycolide acid (Rosomer RG 505, poly-(D,L-lactide-co-glycolide) ester terminated (molecular weight [M_w_] 54,000–69,000); inherent viscosity 0.61–0.74 dl/g; Poly-(vinyl alcohol)—87–90% hydrolyzed, average M_w_ 30,000–70,000 was purchased from Sigma-Aldrich (St. Louis, MO, USA). BS3 (bis(sulfosuccinimidyl)suberate) was purchased from Thermo Fisher Scientific (Waltham, MA, USA). Class B CpG Oligonucleotide-Murine TLR9 ligand (CpG-ODN 1826) was obtained from InvivoGen (San Diego, CA, USA). Rosewell Park Memorial Institute (RPMI) 1640 media, Penicillin-Streptomycin (pen-strep), and fetal bovine serum (FBS) were obtained from Thermo Fisher Scientific (Waltham, MA, USA).

### 2.2. Cell Line and Tumor Antigen Preparation (Tag) Preparation

4T1 cell line was obtained from American Type Culture Collection (ATCC, Manassas, VA, USA), and grown in RPMI 1640 media supplemented with 10% heat-inactivated FBS and 1% pen-strep. Cells were heat-shocked at 42 °C for 1 h to induce immunogenic cell death [19,20] and then allowed to recover at 37 °C with 5% CO_2_ for 2 h. Cells were trypsinized, washed, and resuspended in phosphate-buffered saline (PBS) at a concentration of 2 × 10^8^ cells/mL and then subjected to five cycles of freeze and thaw in liquid nitrogen and a 37 °C water bath. Whole tumor-cell lysate (Tag) was collected as supernatant after centrifugation at 10,000× *g* for 10 min to remove cellular debris. The total protein concentration of Tag was estimated using a Pierce bicinchoninic acid (BCA) assay with albumin as the protein standard (Thermo Scientific, Waltham, MA, USA).

### 2.3. Formulation of Nasal Nano Vaccine (CpG-NP-Tag)

Slight modifications to our earlier reported double-emulsion approach followed by solvent evaporation [17,18] were employed to encapsulate whole cell lysate (Tag) in immune-activating NPs generating CpG-NP-Tag NPs (nasal nano-vaccine) and control formulations (NP-Tag and NP) capable of intranasal immunization. First, NP-Tag NPs were prepared using primary emulsion (w/o) obtained from vortexing 200 μL of Tag solution (25 μg/μL) in 1 mL of the organic phase (PLGA (70 mg) in ethyl acetate (1 mL)) for 1 min. Next, the primary emulsion (w/o) was added to 3 mL aqueous (w) phase (BS3 (0.5 mg/mL) in 2.5% polyvinyl alcohol). The mixture was sonicated on ice using the ultrasonic processor UP200H system (Hielscher Ultrasonics GmbH, Germany) for 1 min to form activated nanoparticles (NPs) (Figure 1). After three washes with distilled water, NPs were resuspended in 5% w/v sucrose solution. Finally, NPs were freeze-dried and lyophilized (under 200 µm vacuum) on the ATR FD 3.0 system and stored at −20 °C until further use. For optimal ligation of CpG, CpG ligand, and resuspended NPs (1:240 w/w ratio) were incubated on an orbital shaker for 45 min at room temperature, and excess ligand was removed with PBS washes to obtain intranasal CpG-NP-Tag.

### 2.4. Characterization of Nanoparticles

Particle size, polydispersity index (PDI), and Zeta potential were measured by resuspending 1 mg of nanoparticles in 1 mL of distilled water to generate a 1:10 dilution from the original concentrated preparation before measuring particle size and Zeta potential using the Zetasizer instrument (Malvern Instruments Ltd., Malvern, UK). The amount of Tag encapsulated in the NPs was calculated based on the amount of protein extracted after degrading a known quantity of NPs (5 mg) by resuspending in 100 mM NaOH + 0.05% SDS (Sigma-Aldrich, St. Louis, MO, USA) followed by incubation at 37 °C on a shaker for 18 h as previously reported [17]. Samples were centrifuged at 11,000× *g* at 4 °C for 10 min, and the supernatants were collected. The protein content of the supernatant was estimated. Encapsulation efficiency was determined using the formula: the amount of protein encapsulated/initial amount of protein used in encapsulation × 100%.

### 2.5. Animal Models

BALB/C female mice (5–6 weeks) were obtained from Charles Rivers Laboratories and maintained at the specific pathogen-free barrier facility at the University of North Texas Health Science Center (UNTHSC). All procedures were performed per the guidelines of the Institutional Animal Care and Use Committee (IACUC) protocol #2021-0039 at UNTHSC.

### 2.6. Intranasal Immunization

The nasal nano-vaccine (Intranasal CpG-NP-Tag), control NP formulations (NP-Tag and NP), and a mixture of CpG and Tag (CpG + Tag) were delivered by intranasal instillation in female BALB/C mice under light anesthesia (Isoflurane) in a 20 μL volume. Each mouse received 6 mg of NPs dissolved in 50 μL PBS. CpG + Tag immunized mice received 25 μg of CpG (the same amount used in CpG-NP-Tag conjugation) and 138 μg of Tag (the exact amount released from 6 mg NP-Tag) in 50 μL of PBS.

### 2.7. Tumor Challenge and Bioluminescence Imaging

For the tumor challenge, 7 days post the third immunization, mice were challenged with 2 × 10^5^ 4T1-luc^+^ cells by intravenous administration via lateral tail vein (Figure 2A). To track the progression of the tumors in the lungs, mice were injected intraperitoneally with 100 μL of D-luciferin (PerkinElmer, Waltham, MA, USA), and whole mouse images were acquired using an IVIS LUMINA III Imaging system (PerkinElmer, Waltham, MA, USA) to visualize and quantify bioluminescence on days 7, 9, and 11 after tumor challenge. To visualize and quantify bioluminescence within lung tissue, mice were injected intraperitoneally with 100 μL of D-luciferin, humanely euthanized 11 days post tumor challenge, and lungs were promptly excised and imaged (IVIS Lumina).

### 2.8. Serum and BALF Collection

Serum was collected by the retro-orbital bleeding technique. Briefly, mice were placed under light anesthesia using isoflurane. Blood was collected by inserting a heparinized Natelson blood collection tube (Fisher Scientific, Waltham, MA, USA) into the venous sinus behind the eyeball. Blood was allowed to clot at room temperature, followed by centrifugation at 10,000× *g* for 10 min. Serum was collected as supernatants and stored at −80 °C until further use.

Mice were humanely euthanized, and BALF was collected by placing mice in dorsal recumbency. PBS (1 mL) containing 100 µM EDTA and 1% protease inhibitor was injected into the lungs of the mice via the trachea. The solution was gently aspirated, and the syringe was removed. The recovered lavage fluid was transferred into a 15 mL conical tube and placed on ice, followed by centrifugation at 200× *g* for 10 min to remove cells and cellular debris. The supernatant was stored at −80 °C until analysis. The remaining BALF cellular fraction was collected and subjected to ACK lysis buffer to remove red blood cells, followed by centrifugation at 300× *g* for 10 min. The cells were then resuspended in FACS buffer (PBS, 2%, FBS, 2 mM disodium EDTA, 2 mM NAN_3_) followed by flow staining with the fluorescent antibody cocktail outlined in the flow cytometry section below for analysis by flow cytometry.

### 2.9. Isolation of Lung and Splenic Leukocytes

Harvested lungs were perfused with 1 mL of sterile PBS and dissected into single lobes using a sterilized razor blade, followed by rinsing with PBS (pH 7.2) in a petri dish to remove excess blood. Lung lobes were digested in gentleMACs C tubes (Miltenyi Biotech, Auburn, CA, USA) containing a digestion buffer lung dissociation kit and processed as a single cell suspension by passing through a 70 μm strainer (Miltenyi Biotech, Auburn, CA, USA). The remaining red blood cells from the lung suspension were lysed with ACK lysis buffer, and the cells were resuspended in RPMI 1640 media supplemented with 10% FBS and 1% pen-strep.

Harvested spleens were mashed through a 40 μm nylon mesh to obtain single-cell suspension (splenocytes). The remaining red blood cells were lysed with ACK lysis buffer, and the cells were resuspended in RPMI media supplemented with 10% FBS and 1% pen-strep.

### 2.10. Flow Cytometry

BALF cells were stained with the fluorochrome antibody cocktail CD3-APC (clone 145-2C11; Tonbo Biosciences, San Diego, CA, USA), CD4-BV605 (clone GK 1.5; Biolegend, San Diego, CA, USA), CD8-BV650 (clone; 53-6.7; Biolegend, San Diego, CA, USA), NK1.1-PerCP (clone PK136; Biolegend, San Diego, CA, USA). Lung leukocytes and splenocytes were washed twice, resuspended in FACS buffer, and stained with the fluorochrome antibody cocktail: CD8-FITC (clone 53-6.7; Biolegend, San Diego, CA, USA,), CD44-BV650 (clone IM7; Biolegend, San Diego, CA, USA), CD62L-PE (clone W18021D; Biolegend, San Diego, CA, USA), To identify lung resident-memory T cells, leukocytes were stained with the following fluorochrome antibodies CD3-APC (clone 145-2C11; Tonbo Biosciences, San Diego, CA, USA), CD4-BV711 (clone RM 4-5; Biolegend, San Diego, CA, USA), CD8-BV650 (clone 53-6.7; Biolegend, San Diego, CA, USA), CD69-APC-CY7 (clone H1.2F3; Tonbo Biosciences, San Diego, CA, USA), CD103-FITC (clone QA17A24; Biolegend, San Diego, CA, USA). To identify activated phenotypes, 1 × 10^6^ leukocytes and splenocytes were co-cultured with 5 × 10^4^ tumor cells in Linbro 24 flat bottom well plates (ICN biomedicals, Ohio, USA) for 6 h. Leukocytes and splenocytes were harvested after culturing and stained with the following fluorochrome antibodies CD3-APC (clone 145-2C11; Tonbo Biosciences, San Diego, CA, USA), CD4-BV711 (clone RM 4-5; Biolegend, San Diego, CA, USA), CD8-FITC (clone 53-6.7; Biolegend, San Diego, CA, USA), CD69-APC-CY7 (clone H1.2F3; Tonbo Biosciences, San Diego, CA, USA) and CD44-BV650 (clone IM7; Biolegend, San Diego, CA, USA). Data were acquired on Cytek Aurora 4-laser flow cytometer (Cytek, Fremont, CA, USA) and analyzed using FlowJo V 10.8.1.

### 2.11. Intracellular Staining

Leukocytes from previously immunized (NP, NP-Tag, CpG + Tag, and CpG-NP-Tag) mice were restimulated with tumor cells. A total of 5 × 10^4^ 4T1 tumor cells were seeded in a flat-bottom 24-well plate overnight, and subsequently, 1 × 10^6^ of leukocytes were added and incubated at 37 °C with 5% CO_2_. After one hour of incubation, Golgi plug (BD biosciences)—the protein transport inhibitor containing brefeldin A, was added. Unstimulated pneumocytes were used as the negative control. After 5 h of incubation, pneumocytes were harvested and suspended in FACS buffer to stain with the fluorochrome antibody cocktail: CD3-APC (clone 145-2C11; Tonbo Biosciences, San Diego, CA), CD4-BV711 (clone RM 4-5; Biolegend, San Diego, CA, USA), CD8-BV650 (clone 53-6.7; Biolegend, San Diego, CA, USA), CD69-APC-CY7 (clone H1.2F3; Tonbo Biosciences, San Diego, CA, USA), CD103-FITC (clone QA17A24; Biolegend, San Diego, CA, USA). For the staining of intracellular granzyme B (GZMB), cells were fixed and permeabilized with BD cytofix/cytoperm (BD biosciences), according to the manufacturer’s instructions, followed by intracellular staining with granzyme B-PE (clone QA18A28; Biolegend, San Diego, CA, USA) and data acquired as outlined in Section 2.10.

### 2.12. ELISA

Leukocytes and splenocytes from previously immunized (NP, NP-Tag, CpG + Tag, and CpG-NP-Tag) mice were cultured in 96-well flat bottom microtiter plates (Fisher Scientific, Waltham, MA, USA) in RPMI 1640 media supplemented with 10% FBS and 1% pen-strep. Cells were stimulated with either Tag (5 µg per well) or NP-Tag (amount delivering 5 µg of Tag per well) in culture media with a final volume of 200 µL/well at a cell concentration of 2 × 10^6^ cells/mL. Anti-mouse CD3/CD28 antibodies (BD Biosciences, Franklin Lakes, NJ, USA) and PBS stimulation were used as positive and negative controls, respectively. Supernatants were collected after 4 days of stimulation at 37 °C with 5% CO_2_ and stored at −80 °C until utilized for the measurement of IFN-γ through ELSA using a mouse IFN-γ uncoated ELISA kit (Invitrogen, Waltham, MA, USA) per the manufacturer’s instructions. IgG and IgA concentrations were detected in serum and BALF supernatants using IgG (Total) mouse Uncoated ELISA Kit (Invitrogen, Waltham, MA, USA) and IgA mouse Uncoated ELISA Kit (Invitrogen, Waltham, MA, USA), respectively.

### 2.13. Statistical Analysis

Statistical analyses were performed as stipulated in figure legends using GraphPad Prism software, v.9.4.0. Results are expressed as mean ± SEM with **** *p* < 0.0001, *** *p* < 0.001, ** *p* < 0.01, * *p* < 0.05 being considered statistically significant.

## 3. Results

### 3.1. Characterization of Intranasal CpG-NP-Tag NPs (Nasal Nano-Vaccine)

NP formulations suitable for targeted delivery to the lung were characterized for particle size, polydispersity index (PDI), Zeta potential, and Tag encapsulation efficiency (Table 1). NPs were found to be slightly larger following tumor antigen encapsulation and CpG conjugation. However, all NP formulations were found to be monodispersed with a lower PDI. Tag-encapsulating nanoparticles had a higher loading capacity for tumor antigens compared to our previous formulations [17,18]. The amount of Tag loaded was calculated based on the protein content of encapsulated whole-cell lysate from 4T1 tumor cells and was found to be 23 μg of protein/mg of NP.

### 3.2. Nasal Nano-Vaccine, CpG-NP-Tag, Reduces Lung Colonization by 4T1 Breast Tumor Cells

The 4T1 tumor cell line is a murine triple-negative breast cancer (TNBC) obtained from BALB/c mice, which represents a class of poorly immunogenic and aggressive tumors associated with low survival rates that are highly resistant to current immunotherapies [21,22]. The lung has been identified as a common site for 4T1 metastasis and recurrence [23,24]. As such, we tested the ability of our engineered nasal nano-vaccine to prevent 4T1 tumor cell lung colonization as a correlate to lung metastasis in syngeneic mice tumor models (Figure 2A). The bioluminescence signal intensity detected on days 7, 9, and 11 after 4T1-luc^+^ cell challenge was significantly (*p* ≤ 0.05) reduced in groups immunized with CpG-NP-Tag and CpG + Tag compared with NP-Tag and NP immunized mice. CpG-NP-Tag-immunized mice had the lowest bioluminescence signal intensity, hence the smallest tumor burden visualized by both whole mouse imaging and ex vivo lung imaging (Figure 2B,C).

### 3.3. Nasal Nano-Vaccination Promotes Cellular and Humoral Immune Responses in the Lungs of Mice Challenged with 4T1 Tumor Cells

CD8^+^, CD4^+^ T cells, and NK cells are known to mediate tumor immunosurveillance in the lungs against invading tumor cells [25]. Therefore, we assessed the accumulation of CD8^+^, CD4^+^ T cells, and NK cells and the concentrations of their effector molecules (IFN-γ and granzyme B) in the bronchioalveolar lavage fluid (BALF) of previously immunized mice that were subsequently challenged with 4T1 tumor cells. Analysis of cells collected from BALF using fluorescence-activated cell sorting (FACS) revealed a significant (*p* ≤ 0.05) increase in CD8^+^ and CD4^+^ T cells but not NK cells of CpG-NP-Tag-immunized groups compared to all immunized treatment groups (Figure 3A). By comparison, the CpG-NP-Tag vaccination did not significantly alter the number of CD8^+^ and CD4^+^ T cells in the spleen (Figure 3B). Further differentiation of CD8^+^ and CD4^+^ T cells revealed a significant (*p* ≤ 0.05) increase in the activated phenotype, CD3^+^ CD69^+^ CD8^+^ and CD3^+^ CD69^+^ CD4^+^ T cells in the BALF of CpG-NP-Tag-immunized mice (Figure 3C). By contrast, no significant difference in the frequency of CD3^+^ CD69^+^ CD8^+^ and CD3^+^ CD69^+^ CD4^+^ T cells of CpG-NP-Tag-immunized mice were observed in the spleen (Figure 3D).

The increased infiltration of both CD3^+^ CD69^+^ CD8^+^ and CD3^+^ CD69^+^ CD4^+^ T cells in BALF was accompanied by a significant (*p* ≤ 0.05) increase in IFN-γ and increased granzyme B (approaching significance) (Figure 3E). We also assessed the effect of intranasal immunization on humoral antibody responses in BALF during the tumor challenge. Both CpG-NP-Tag- and CpG + Tag-immunized mice had increased concentrations of BALF IgA, albeit not significantly different. In addition, we determined the levels of serum IgG responses to intranasal immunization. A significant (*p* ≤ 0.05) increase in serum IgG concentrations was found in CpG-NP-Tag- and CpG + Tag- compared to NP-Tag- and NP-immunized mice, but not IgA.

### 3.4. Intranasal Nano-Vaccine Induces Tumor-Specific Cell-Mediated and Humoral Antibody Immune Responses

Given the ability of the nasal nano-vaccine to reduce lung colonization by 4T1 tumor cells, we were interested in identifying immune-based mechanisms mediated by nasal nano-vaccination to confer protection. Initially, we determined the quality and phenotype of immune responses induced by nasal nano-vaccination over time (Figure 4).

Owing to the heightened T-cell-mediated immune response against 4T1 tumor cell challenge in nasal nano-vaccine-immunized mice as detected in the BALF, we assessed the ability of the nasal nano-vaccine to induce T cell responses in the absence of tumor burden. We determined the frequencies of Tag-experienced T cells in the form of effector memory (CD44^+^CD62L^−^) (T_EM_) and Tag-inexperienced T cells as naïve (CD44^−^CD62L^+^) T cells in the lungs and spleen. A significant (*p ≤* 0.05) increase in the frequency of lung CD8^+^ T_EM_ cells was observed in CpG-Tag-NP-immunized mice on days 7 and 21 days at the expense of naïve T cells but not observed by splenocytes on days 7 and 21 (Figure 5A; Appendix A). The frequency of CD4^+^ T_EM_ was also shown to be significantly (*p ≤* 0.05) increased in the lungs at the expense of naïve T cells but not in the spleen of nasal nano-vaccine immunized mice (Figure 5B; Appendix A).

We also observed both nasal nano-vaccine and CpG + Tag immunization to significantly (*p ≤* 0.05) increase concentrations of BALF IgA at both 7 days and 21 days post immunization (Figure 5C). In addition, serum IgG was significantly (*p ≤* 0.05) increased in CpG-Tag-NP-immunized mice day 7 post immunization and only increased in CpG + Tag-immunized mice 21 days post immunization compared to NP-immunized mice (Figure 5D).

To determine if T_EM_ cells in the lungs of nasal nano-vaccine-immunized mice were tumor-specific, we investigated the ex vivo responses of lung and spleen cells from nasal nano-immunized mice at 21 days post immunization. Following an ex vivo 6-h co-culture of lung and spleen cells with viable 4T1 tumor cells, we observed a significantly (*p ≤* 0.05) higher proportion of both CD8^+^ CD69^+^ CD44^hi^ and CD4^+^ CD69^+^ CD44^hi^ T cells from the lungs but not spleen (Figure 6A,B).

We then determined whether the increased proportions of CD8^+^ CD69^+^ CD44^hi^ and CD4^+^ CD69^+^ CD44^hi^ T cells following ex vivo restimulation with viable 4T1 tumor cells would correspond with increased IFN-γ production. Lung leukocytes but not splenocytes isolated from nasal nano-vaccine-immunized mice produced significantly (*p ≤* 0.05) higher IFN-γ upon restimulation with tumor antigen-encapsulating nanoparticles (NP-Tag) in contrast to unencapsulated tumor antigen (Tag). No significant differences in IFN-γ production were found in splenocyte cultures (Figure 6C).

### 3.5. Intranasal Nano-Vaccine Induces Preferential Expansion of Lung CD103^+^ CD69^+^ Resident Memory T Cells

Recent evidence supports that some effector T cells do not recirculate but maintain tissue residency and can persist lifelong in tissues after clearing antigens [26,27]. Therefore, we compared the induction of lung resident-memory (CD103^+^ CD69^+^) T (T_RM_) cells to their circulating (CD103^−^) counterparts (T_CIR_ cells). We showed that intranasal CpG-NP-Tag administration produces a significant (*p* ≤ 0.05) accumulation of CD103^+^ CD69^+^ CD8^+^ T cells compared to CD8^+^ T_CIR_ cells in lungs of CpG-NP-Tag-immunized mice at both days 7- and 21-days post immunization (Figure 7A). Similarly, we also observed an increased accumulation of CD103^+^ CD69^+^ CD4^+^ T cells in the lung compared to T_CIR_ cells of CpG-NP-Tag-immunized mice at both days 7- and 21-days post immunization (Figure 7B; Appendix A). To further verify if the accumulation of CD8^+^ T_RM_ was due to intranasal CpG-NP-Tag immunization, we isolated lung leukocytes 21 days post immunization from mice and stimulated ex vivo with 4T1 tumor cells. Using FACS analysis, we show a higher proportion of CD8^+^ T_RM_ cells from nano-vaccine-immunized mice with higher effector function by producing granzyme B as compared to CD8^+^ T_CIR_ cells (Figure 7C,D).

## 4. Discussion

Most cancer vaccines rely on dendritic cells to activate tumor antigen-specific CD4^+^ and CD8^+^ T cells, as well as antibody-producing B cells capable of long-lasting antitumor responses through the generation of immunological memory [2,28,29,30,31]. However, cancer antigens constituting self-molecules are often inadequate at inducing dendritic cell activation and maturation [32,33]. Consequently, ligands, usually non-self-molecules, fulfill this requirement by binding to pattern recognition receptors (PRR) as adjuvants in developing cancer vaccines. To this end, our laboratory has focused on engineering a PLGA polymer-based immune-activating nanoparticle functionalized with the bacterial component, CpG, in which its ligand Toll-Like receptor (TLR)-9 is expressed by antigen-presenting cells (APCs). TLR-9 is located in endosomal compartments of APCs, affording nanoparticles to be readily trafficked into endosomes following cellular uptake, where they can be degraded to release encapsulated tumor antigens for processing, presentation on MHC molecules, and subsequent activation of tumor antigen-specific T cells [18]. We have demonstrated that encapsulated tumor antigens (Tag) made up of purified membrane proteins from murine 4T1 tumor cells to generate immune-activating nanoparticles (CpG-NP-Tag) reduced primary tumor size and tumor weight associated with an accumulation of CD4^+^ and CD8^+^ T cells as well as the detection of IFN-γ within the solid tumor mass [17]. To enhance the translation feasibility for a cancer vaccine for the prevention of metastatic disease, we propose a prophylactic approach to prevent metastasis following a primary tumor, where patients’ primary tumors can serve as a source of antigens. As a first step in conceptualizing this model, the current study tested the ability of intranasal immunization of CpG-NP-Tag to protect against experimental colonization of 4T1 breast tumor cells in the lungs as a model of lung metastasis. We formulated a CpG-NP-Tag capable of intranasal administration by increasing its Tag-loading capacity compared to our previous formulations (Figure 1) [17,18]. This allowed for the delivery of small amounts of NP suitable for intranasal immunization yet delivering enough Tag and vaccine adjuvant (CpG) to provoke an immune response. Our results underscore the need for the use of nanoparticles such as PLGA to deliver cancer antigens and adjuvants as they can deliver up to 1000-fold greater into APCs than native antigens leading to robust and lasting immune responses against tumor antigens [34].

Compared to mice that received the vaccine in its native form (CpG + Tag), nasal nano-vaccinated (CpG-NP-Tag) mice had superior protection against breast tumor lung-colonization (Figure 2B,C). This superior protection was attributed to the induction of prominent tumor-reactive CD8^+^, and CD4^+^ T-cell responses as detected in BALF and the lung parenchyma of CpG-NP-Tag-immunized mice. CpG-Tag-NP immunization led to increased accumulation of CD8^+^ and CD4^+^ T cells expressing the activation marker CD69 in BALF when immunized mice were challenged with 4T1 tumor cells via tail vein injection (Figure 3C,D). In addition, lung leukocytes from CpG-NP-Tag had a more significant percentage activated (CD69^+^CD44^hi^ phenotype) and produced more IFN-γ following ex vivo restimulation with viable 4T1 tumor cells and NP-Tag, respectively (Figure 6A–C). APC can only activate CD8^+^ T cells against exogenous antigens by the cross-presentation of antigens on MHC class I molecules [35]. Hence, our findings suggest that using PLGA as a vehicle to co-deliver tumor antigens (Tag) and adjuvant (CpG) activates dendritic cells with the ability for cross-priming tumor-reactive CD8^+^ T-cell responses via MHC class I pathways. At the same time, they maintain their ability to process exogenous antigens via MHC class II pathways to activate CD4^+^ T cells. CpG-NP-Tag- and CpG + Tag-immunized mice demonstrated increased humoral-antibody immune responses by producing IgA and IgG as detected in BALF and serum, respectively (Figure 5C,D). This, together with the observation that, unlike lung CD8^+^ T_EM_, there was no significant difference in lung CD4^+^ T_EM_ between CpG-NP-Tag- and CpG + Tag-immunized mice (Figure 5A,B), suggests that CpG + Tag vaccination could not cross prime tumor-reactive CD8^+^ T cells but promoted CD4^+^ T cells that could predominately be T-follicular helper (TFH) CD4^+^ T cells that are known to provide help to B cells [36] to produce IgA and IgG which protected tumor cell colonization in CpG + Tag-immunized mice compared to NP-Tag- and NP-immunized mice.

The main advantage of the direct inoculation of a cancer vaccine to metastatic sites such as the lung is to induce tissue-resident memory T cells, as increasing evidence supports that accumulation of tissue-resident memory T cells in tumors correlates with a favorable prognosis [37,38,39,40,41,42,43,44]. Tissue-resident memory T cells are a subset of long-lived memory T cells that reside permanently in tissues and do not recirculate. Because they are long-lived and can provide long-lasting antitumor immune responses, their induction by cancer vaccines is quickly becoming a hallmark in determining the efficacy of immune-based cancer vaccines [38,45]. In the lung, these T cells express CD69 and the αE integrin CD103, which binds to the E-cadherin of lung-tissue epithelial cells, thereby supporting lung retention [46,47]. In our study, we demonstrated that intranasal CpG-NP-Tag induced both CD8^+^ and CD4^+^ T_RM_ in the lungs (Figure 7A,B), which were shown to be tumor antigen-specific as CD8^+^ T_RM_ isolated from the lungs of CpG-NP-Tag-immunized mice produced granzyme B following ex vivo restimulation with 4T1 tumor cells from which Tag was derived (Figure 7C,D). The induction of tumor antigen-specific CD8^+^ and CD4^+^ T_RM_ cells prior to the seeding of tumor cells from the lungs by extra-pulmonary tumors is necessary to provide a potent immunocompetent environment to eliminate DTCs thereby preventing their outgrowth from becoming overt metastatic lesions. This can be achieved via a vaccination strategy, as shown in this study.

Avoiding the provocation of systemic inflammatory responses in patients with cancer, especially those with co-morbid autoimmune conditions, is important to immune-based cancer vaccine development [48]. Therefore, we evaluated the effect of intranasal CpG-NP-Tag administration in inducing systemic immune responses by assessing tumor antigen-specific immune responses in the spleen of immunized mice. Our results show no difference in vaccine-elicited T cells across treatment groups because they had similar proportions of effector and naïve T cells in their spleens which did not respond to ex vivo restimulation by 4T1 tumor cells as opposed to T cells from the lungs. In addition, there was no difference in IFN-γ concentrations in the supernatants of splenocyte cultures across treatment groups after stimulation with NP-Tag as opposed to supernatants from cultured lung leukocytes (Figure 6C). Such findings are relevant in developing immune-activating vaccines as the off-target inflammatory response can prove counteractive to preventative measures.

The vaccination strategy reported here compares to in situ vaccination strategies where vaccines and adjuvants are directly injected into the primary tumor [49,50,51]. However, the immunosuppressive microenvironment at the primary tumor site which is not readily available at metastatic sites dampens vaccination-generated immune responses necessitating accompanying strategies to improve in situ vaccination efficacy. By implication, inoculation at metastatic sites is an economical way of improving cancer vaccine effectiveness and reduces the risk of developing metastatic disease.

Previous studies that have investigated intranasal and other pulmonary delivery for cancer vaccines have tested their efficacy against pulmonary tumors and not against tumors that originate elsewhere in the body seeking to metastasize to the lungs [38,52,53,54]. Here, we have shown the protective efficacy of the intranasal cancer vaccine (CpG-NP-Tag) against lung colonization by breast tumor cells mediated by cross-primed tumor-reactive CD8^+^ T-cell responses in the lungs. However, assessing if intranasal CpG-NP-Tag can prevent metastasis to other organs such as the brain, bone, and liver, which are other frequent sites of metastasis too, or if they require direct inoculation with our engineered vaccine was outside the scope for this study.

## Figures and Tables

**Figure 1 pharmaceutics-15-00445-f001:**
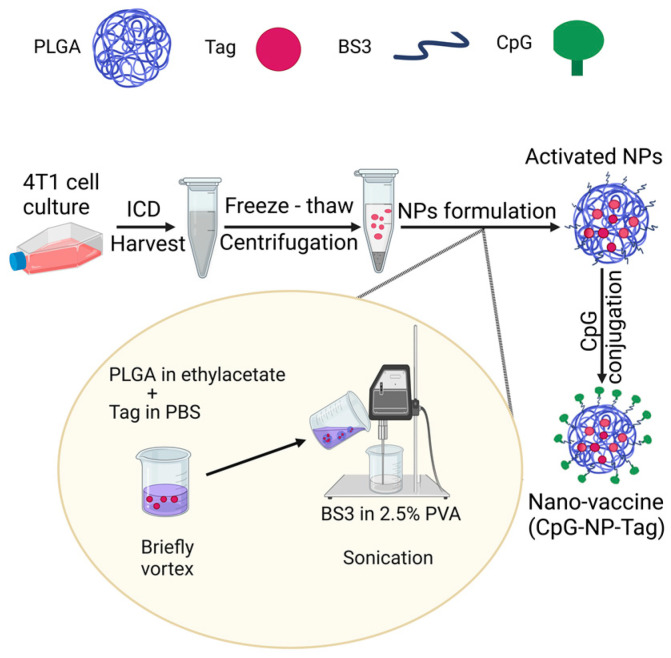
Formulation of nasal nano-vaccine (Intranasal CpG-NP-Tag). Diagram summarizing the main steps involved in the synthesis of the nasal nano-vaccine (Intranasal CpG-NP-Tag). PLGA: poly—(lactide-co-glycolic acid); Tag: Tumor antigen; BS3: Bis(sulfosuccinimidyl)suberate; CpG: Cytosine-phosphate-guanosine oligodeoxynucleotide; NP: Nanoparticle.

**Figure 2 pharmaceutics-15-00445-f002:**
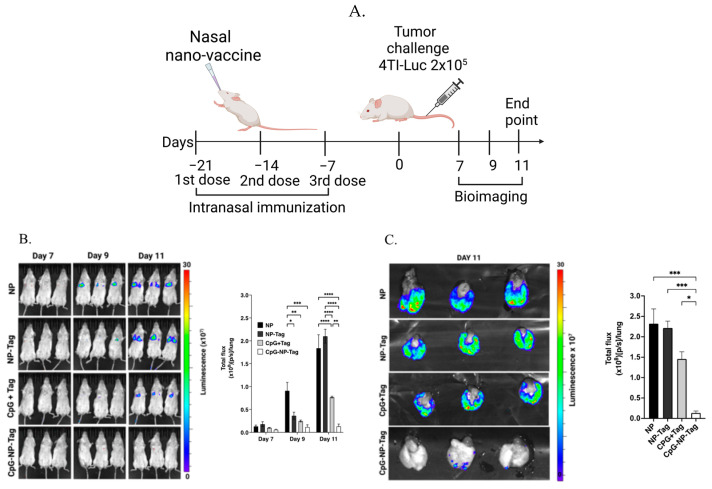
Nasal nano-vaccine immunization increases protection against experimental lung metastasis of 4T1 breast tumor cells. (**A**) Schematic of the study timeline for vaccine dosing followed by tumor challenge and bioluminescence imaging in BALB/C mice. (**B**) Whole mouse bioluminescence images (left) and quantification of luminescence signal intensity (right) 7-, 9-, and 11-days post tumor challenge. (**C**) Excised lungs’ bioluminescence images (left) and quantification of luminescence signal intensity (right) day 11 post tumor challenge. One-way ANOVA followed by Tukey’s multiple comparison tests. **** *p* < 0.0001, *** *p* < 0.001, ** *p* < 0.01, * *p* < 0.05.

**Figure 3 pharmaceutics-15-00445-f003:**
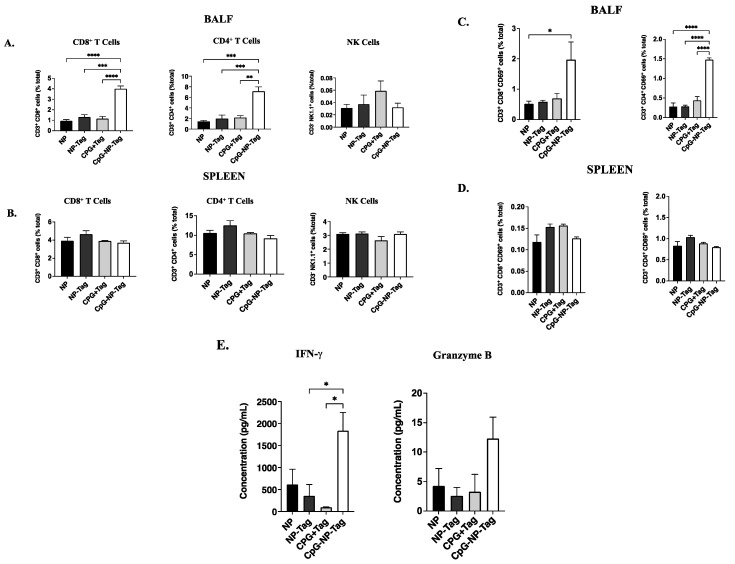
Nasal nano-vaccine immunization increases T-cell immune responses in the lower respiratory after breast tumor challenge via tail vein injection. (**A**,**B**) Frequency of CD8^+^ T cells, CD4^+^ T cells, and NK cells among total cells in the BALF (**A**) and spleen (**B**). (**C**,**D**) Frequency of CD69^+^ among CD8^+^ (right) and CD4^+^ (left) T cells in BALF (**C**) and spleen (**D**). (**E**) Concentrations of IFN-*γ* (left) and granzyme B (right) in BALF collected from experimental mice day 11 after tumor challenge. One-way ANOVA followed by Tukey’s multiple comparison test. **** *p* < 0.0001, *** *p* < 0.001, ** *p* < 0.01, * *p* < 0.05.

**Figure 4 pharmaceutics-15-00445-f004:**
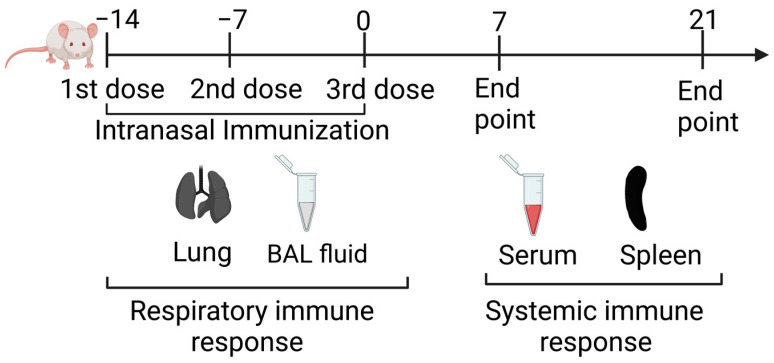
Schematic of the study timeline for vaccine dosing to investigate the time course of immune responses to nasal-nano vaccine in BALB/C mice. Two groups of female BALB/C mice (*n* = 3) were immunized 3 times across their nasal nares with the respective nanoparticles 7 days apart. Mice were then humanely euthanized days 7 and 21 post immunization for first and second group, respectively. Lungs, spleen, BALF, and serum were harvested upon sacrifice to investigate immune responses to vaccine.

**Figure 5 pharmaceutics-15-00445-f005:**
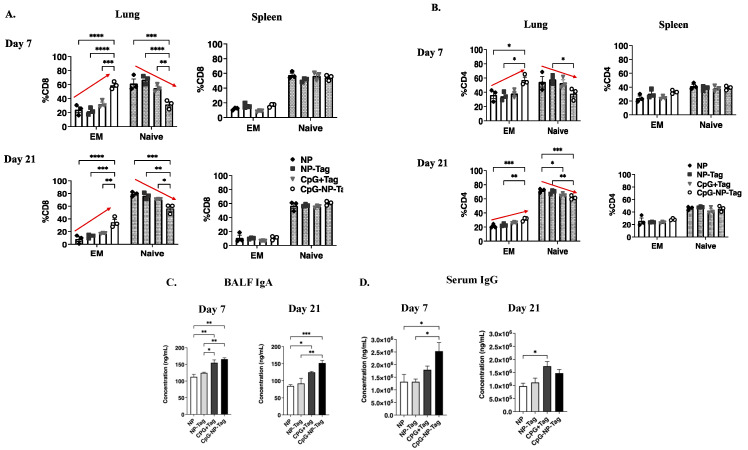
Intranasal CpG-NP-Tag administration increases T-cell and humoral antibody-mediated immune responses in lower respiratory of BALB/C mice. (**A**,**B**) Proportions of T_EM_ (CD44^+^ and CD62L^-^) and naïve (CD44^-^ and CD62L^+^) cells among CD8^+^ (**A**) and CD4^+^ (**B**) T cells in lungs and spleen day 7 and day 21 post immunization. Two-way ANOVA followed by Tukey’s multiple comparison test. (**C**,**D**) Concentrations of BALF IgA (**C**) and serum IgG (**D**) day 7 and 21 post immunization. One-way ANOVA followed by Tukey’s multiple comparison test. **** *p* < 0.0001, *** *p* < 0.001, ** *p* < 0.01, * *p* < 0.05.

**Figure 6 pharmaceutics-15-00445-f006:**
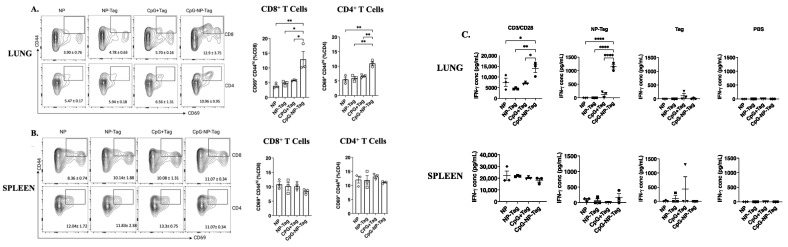
Intranasal CpG-NP-Tag drives tumor antigen-induced T cell activation in lungs of BALB/C mice. (**A**,**B**) Representative FACS blots of CD69 and CD44 expression and proportions of CD69^+^ CD44^hi^ cells within CD8^+^ and CD4^+^ in the lungs (**A**) and spleen (**B**) of experimental mice 21 days post immunization after ex vivo restimulation with viable 4T1 tumor cells. (**C**) Concentrations of IFN-γ in culture supernatants of lung leukocytes (upper panels) and spleenocytes (lower panels) from experimental mice 21 days post immunization after ex vivo stimulation with NP-Tag and Tag for 4 days using CD3/CD28 and PBS as positive and negative controls, respectively. One-way ANOVA followed by Tukey’s multiple comparison test. **** *p* < 0.0001, ** *p* < 0.01, * *p* < 0.05.

**Figure 7 pharmaceutics-15-00445-f007:**
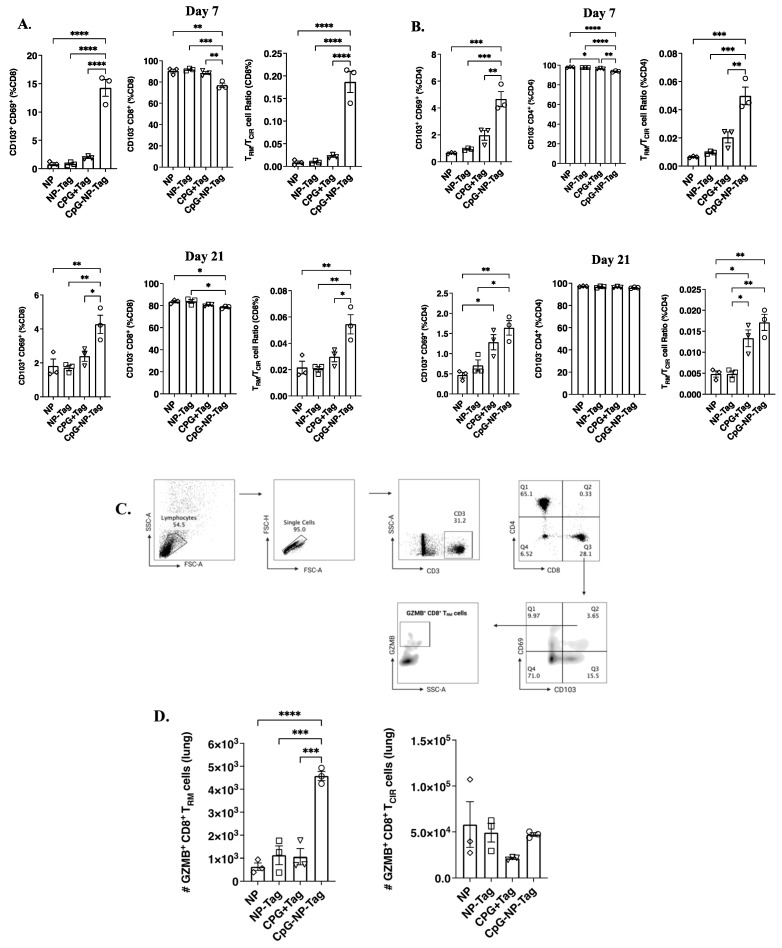
Intranasal CpG-NP-Tag increases accumulation of tumor specific lung resident memory T cells in lungs of Balb/C mice. (**A**,**B**) Proportions of T_RM_ (CD103^+^ and CD69^+^); left, T_CIR_ (CD103^−^ and CD8^+^); middle and ratio of T_RM_/T_CIR_; right among CD8^+^ (A) and CD4^+^ (B) T cells day 7 and 21 post immunization. (**C**) Gating strategy to identify granzyme producing cells within CD8^+^ T_RM_. (**D**) Number of granzyme producing CD8^+^ T_RM_ (left) and CD8^+^ T_CIR_ cells (right) after ex vivo restimulation with viable 4T1 cells. One-way ANOVA followed by Tukey’s multiple comparison test. **** *p* < 0.0001, *** *p* < 0.001, ** *p* < 0.01, * *p* < 0.05.

**Table 1 pharmaceutics-15-00445-t001:** Physicochemical characterizations of NP formulations.

Sample	Particle Size (nm ± SD)	PDI	Zeta Potential (mV ± SD)	Encapsulation Efficiency (%)	µg Protein/mg NP
CpG-NP-Tag	259.0 ± 2.10	0.138	−14.56 ± 0.23	51 ± 7.70	23
NP-Tag	255.8 ± 1.84	0.248	−6.98 ± 0.18	51 ± 7.70	23
NP	219.0 ± 0.82	0.040	−5.52 ± 0.19	-	-

NP: Nanoparticle; PDI: Polydispersity Index; SD: Standard deviation. Physicochemical characterization of CpG-NP-Tag and control (NP-Tag and NP) formulations: The NPs were characterized using previously reported protocol [17] by dynamic light scattering and Zeta potential measurements. The data shown in Table 1 are representatives of the mean value ± SD of readings from three different batches.

## Data Availability

The data presented in this study are available in the Appendix A.

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
