# Peer review of "Nasal Tumor Vaccination Protects against Lung Tumor Development by Induction of Resident Effector and Memory Anti-Tumor Immune Responses"

_pharmaceutics, 2023, doi:10.3390/pharmaceutics15020445_

Round 1

Reviewer 1 Report

This manuscript described a nasal nano-vaccine system based on the CpG coated tumor antigen (Tag) encapsulated PLGA nanoparticles (CpG-NP-Tag). Results showed that CpG-NP-Tag induced significant mucosal immune responses reflected by the increased expression of T cells, IFN-γ and granzyme B. However, several questions, including those below, should be addressed before considering its publication in Pharmaceutics.

1.        The information/data related to the success of CpG-NP-Tag delivery across the respiratory tract after its intranasal administration should be provided.

2.        What about the biodistribution of CpG-NP-Tag?

3.        The antigens originating from the real tumors are different with those from cancer cells cultured in a dish. Therefore I am curious that why the authors did not use tumors to generate Tag, but used the 2D cultured cells instead.

4.        In absence of pre-metastatic niche, does it make sense that the method of intravenous injection of 4T1 cells via tail vein can still mimic the real lung metastasis?

5.        From the immunization data, there was little difference between NP and NP-Tag groups, while CpG-NP-Tag elicited a dramatic enhancement of immune responses compared to NP-Tag group. Therefore it seems that CpG adjuvant, not the Tag, takes the leading role in vaccination. Should authors comment on that? Furthermore, to illustrate both Tag and CpG are vital constitutes in generating effective antitumor immune responses, it is better to include the formulation of CpG-NP without Tag, which is unfortunately lacking in the study.

6.        Can the authors provide some direct proofs regarding to the inhibition of lung metastasis by CpG-NP-Tag?

7.        In the Discussion Section, a comparison between the current study with previously published strategies based on the in situ tumor vaccination (e.g. Nat. Med. 2019, 25, 814; Small. 2022, 18, 2200993; Adv. Mater. 2021, 33, 2100628; Biomaterials 2021, 275, 120921; Chin. Chem. Lett. 2021, 32, 1770) is suggested.

Author Response

Comments from Reviewer 1

Thank you for your comments. We believe addressing your comments has help us improve the research presented within as well as facilitated our future studies.

Comment 1: The information/data related to the success of CpG-NP-Tag delivery across the respiratory tract after its intranasal administration should be provided.

Response: Thank you for your comment. We have previously demonstrated and published data (referenced below) on the success of intranasal delivery and lung retention of our NP based vaccine encapsulating a bacterium antigen. This is consistent with what we have reported, as we demonstrated the success of CpG-NP-Tag delivery across the respiratory tract by showing in our manuscript that intranasal CpG-Tag-NP compared to NP immunized mice induces a strong mucosal immune response because of increase in BALF IgA and lung T-cell responses against Tag. Again, we could only detect induction of T-cell immune response in the lungs and not spleen of immunized mice, indicating the vaccine was retained and elicited its effects in the lung.

Reference: Mott, B., Thamake, S., Vishwanatha, J. et al. Intranasal delivery of nanoparticle-based vaccine increases protection against S. pneumoniae. J Nanopart Res 15, 1646 (2013). https://doi.org/10.1007/s11051-013-1646-x

Comment 2: What about the biodistribution of CpG-NP-Tag?

Response: Thank you for your comment. As we have pointed out in our response to comment 1 above, the referenced manuscript earlier published showed the retention of the NP based vaccine in the lung. Even so, we have demonstrated in our manuscript the absence of any immune response in the spleen which indicates that any CpG-NP-Tag found in systemic circulation was not significant as opposed to the lung where it elicited its effects.

Comment 3: The antigens originating from the real tumors are different with those from cancer cells cultured in a dish. Therefore, I am curious that why the authors did not use tumors to generate Tag but used the 2D cultured cells instead.

Response: You have raised an important point here. However, there is not much difference between antigens originating from a cancer cell line and established tumors (referenced below). Again, established tumors turn on programming to secrete lots of immunosuppressive proteins and cytokines. They also recruit lots of suppressor immune cells (such as myeloid suppressor cells) into the tumor microenvironment. Therefore, using established tissues will include the high probability of having all kinds of immunosuppressive proteins and cytokines as part of extracted tumor antigens which will dampen the immune response (for example inhibit dendritic cell maturation). The use of the cell line allows us to extract proteins (potentially antigens) that are exclusively expressed by the tumor and avoid the presence of proteins and cytokines released by other cells in the microenvironment of established tumors.

Reference: Kloudová K, Hromádková H, Partlová S, Brtnický T, Rob L, Bartůňková J, Hensler M, Halaška MJ, Špíšek R, Fialová A. Expression of tumor antigens on primary ovarian cancer cells compared to established ovarian cancer cell lines. Oncotarget. 2016 Jul 19;7(29):46120-46126. doi: 10.18632/oncotarget.10028. PMID: 27323861; PMCID: PMC5216785.

Comment 4:  In absence of pre-metastatic niche, does it make sense that the method of intravenous injection of 4T1 cells via tail vein can still mimic the real lung metastasis?

Response: Again, this an important point you have raised, however, the lateral tail vein injection method (intravenous injection) also known as the experimental metastasis model is a mainstay of metastasis research for most primary tumors that metastasize to the lungs (referenced below). As you have pointed out, its limitation is that it evaluates lung metastasis without a primary tumor and therefore ignores the biologic progression from primary tumor to metastatic lesion in the lung. With this limitation of tail vein injection in mind, in our manuscript we tested the efficacy of our vaccine at preventing the colonization of the lung by breast tumor cells that disseminate to the lung. Colonization of the lung by disseminated tumor cells is only the 4th step of the lung metastatic cascade and we mimicked that by lateral tail vein injection that sends tumor cell to the lung. Hence the conclusion from our paper is that, by preventing lung colonization by 4T1 tumor cells through intranasal immunization with our vaccine, lung metastasis can be halted. Further studies are being conducted to evaluate the efficacy of the vaccine in the context of an existing primary tumor.

Reference: Thies KA, Steck S, Knoblaugh SE, Sizemore ST. Pathological Analysis of Lung Metastasis Following Lateral Tail-Vein Injection of Tumor Cells. J Vis Exp. 2020 May 20;(159). doi: 10.3791/61270. Erratum in: J Vis Exp. 2020 Dec 2;(166): PMID: 32510518.

Comment 5:  From the immunization data, there was little difference between NP and NP-Tag groups, while CpG-NP-Tag elicited a dramatic enhancement of immune responses compared to NP-Tag group. Therefore, it seems that CpG adjuvant, not the Tag, takes the leading role in vaccination. Should authors comment on that? Furthermore, to illustrate both Tag and CpG are vital constitutes in generating effective antitumor immune responses, it is better to include the formulation of CpG-NP without Tag, which is unfortunately lacking in the study.

Response: Thank you for the comment. That is a great observation, and it is consistent with what we expected.  As you rightly pointed out concerning CpG adjuvant, when using whole cell lysate, the use of vaccine adjuvant is very important to initiate the required inflammation needed to elicit a T-cell immune response against antigens in the whole cell lysate. Most antigens in the whole cell lysate are tumor associated antigens (usually self-proteins) that cannot induce the maturation of dendritic cells by themselves (dendritic cells are activated and mature by recognizing non-self-molecules such as CpG).  The use of CpG alone can induce inflammation but cannot elicit a T-cell response because T-cells only get activated by recognizing antigens in the form of peptides presented by MHC molecules. CpG is not a peptide, and it has not been shown by any study to generate a T-cell response on its own, except inducing T-cell immune response against peptide antigens.  Our data proves that both CpG and Tag are necessary requirements for T-cell immune responses against T-cell responses generated against Tag. CpG induces dendritic cell maturation, Tag provides the peptides to activate the T-cells. On the basis of this immunological knowledge, we did not include CpG-NP in our control because of we administered the vaccine prior to tumor challenge. In the presence of an existing tumor, perhaps CpG-NP can be an ideal control, because it will induce the maturation of dendritic cells to pick up dying tumor cells to present to T-cell leading to antitumor T-cell immune response as demonstrated by most in-situ vaccinations against cancer. We used a different vaccination protocol in our manuscript by investigating the prophylactic use of cancer vaccine when giving prior to tumor challenge. In summary, our results showed the anti-tumor immune response generated was predominately a T-cell immune response. CpG cannot not produce this immune response on its own in the absence of a tumor antigen.

Comment 6: Can the authors provide some direct proofs regarding to the inhibition of lung metastasis by CpG-NP-Tag?

Response: Thank you for the comment. We have shown in our manuscript that CpG-NP-Tag prevents lung colonization by 4T1 tumor cells. The lung metastatic cascade is a multistep process involving 1. The detachment of tumor cells from the primary tumor 2. The intravasation of the tumor cell into circulation. 3. Extravasation of tumor cells into the lung 4. The outgrowth of the disseminated tumor cell into micro metastasis eventually leading to overt metastasis. The work reported in this paper is a potential to prevent lung metastasis because it prevents the outgrowth of tumor cells that disseminate to the lung (experimentally demonstrated using tail vein injection) from colonizing the lung. Therefore, by preventing step 4, we hope to halt lung metastasis in its track.

Comment 7: In the Discussion Section, a comparison between the current study with previously published strategies based on the in situ tumor vaccination (e.g. Nat. Med. 2019, 25, 814; Small. 2022, 18, 2200993; Adv. Mater. 2021, 33, 2100628; Biomaterials 2021, 275, 120921; Chin. Chem. Lett. 2021, 32, 1770) is suggested.

Response: Thank you for this suggestion. We believe this will improve our paper and has been duly added. We have added as indicated in quotation below to our discussion.

“The vaccination strategy reported in here compares to in-situ vaccination strategies where vaccines and adjuvants are directly injected into the primary tumor[50-52]. However, the immunosuppressive microenvironment at the primary tumor site which is not readily available at metastatic sites dampens vaccination generated immune responses necessitating accompanying strategies to improve in-situ vaccination efficacy. By implication, inoculation at metastatic is an economical way of improving cancer vaccine effectiveness and reduces the risk of developing metastatic disease.”

Reviewer 2 Report

In this manuscript, the authors proposed a new strategy to induct resident effector and memory anti-tumor immune responses by developing innovative nasal tumor vaccination. The engineering nano-vaccine (CpG-NP-Tag) induced T cell-mediated mucosal immune response via dendritic cell activation and maturation, followed by long-lasting antitumor responses. These findings indicate potential to prevent overt lung metastases from existing extra pulmonary tumors by engineering nasal nano-vaccine. Meanwhile, the manuscript was presented in a logic manner, and also well written. Thus, I recommend the acceptance in the journal after minor revisions.

Some specific issues should be addressed as follows:

1.     More experiments should be provided to confirm the biosafety of CpG-NP-Tag such as histopathological analysis.

2.     The dendritic cell uptake and intracellular localization of CpG-NP-Tag should be analyzed using fluorescence imaging.

3.     The morphology and dispersity of nano-vaccine should be detected by SEM.

4.     The CpG loading efficiency of nano-vaccine should be characterized using DNA gel electrophoresis.

5.     The resolution of schematics and figures should be improved (e.g., figure 4A and 6A).

6. The manuscript should be carefully checked to avoid the spelling, expression and grammar mistakes, and proofread by a native professor. For instance:

(1) In Figure 1, “4TI”?

(2) In line 589, “IFN-” in the discussion;

7. For a better presentation, some figures and literal representation needed to be revised.

(1) The fonts of texts should be consistent (e.g., Figure 3a and 3b);

(2) There should be space between word and unit (e.g., line 170, 11000g and 220, 2mM);

(3) In Figure 2C, “AY” should be written in lower case.

Author Response

Comments from Reviewer 2

Thank you for your comments. We believe your comments have improved our research presented within as well as facilitated our future studies.

Comment 1:  More experiments should be provided to confirm the biosafety of CpG-NP-Tag such as histopathological analysis.

Response: Thank you for the comment. We have included data on CpG toxicity (effect on mice weight) in our supplementary data. Again, evidence from pre-clinical and clinical studies indicates that CpGs are reasonably safe when administered as vaccine adjuvants. In our study we did not exceed the maximum dose at which CpG is considered toxic (exceeding 25 µg per mouse). To ensure that CpG-NP-Tag is safe we monitored the weights of mice (normally used to access CpG toxicity when used as vaccine adjuvant) after intranasal immunization. We noticed small reduction in mice weights after first dose of CpG-NP-Tag, which is expected because CpG induces potent inflammation by mimicking bacterial infections. The mice did not lose weight equivalent to 25% of their initial body weight which is usually associated with CpG toxicities when given at high doses. Again the 2nd and 3rd doses of CpG-NP-Tag was well tolerated because it was not associated with any weight lost (mice had regained weight lost from 1st dose of CpG administration). Hence CpG-NP-Tag is considered to be safe. Data has been included in supplementary data.

Comment 2: The dendritic cell uptake and intracellular localization of CpG-NP-Tag should be analyzed using fluorescence imaging.

Response: Thank you for the comment. We have previously reported this in a published manuscript when we proved the immunogenicity of CpG-NP-Tag. We show that CpG-NP-Tag is preferentially taken up by dendritic cells and it traffics to the endosomes as compared to our NP controls. This initial study provided the basis for the use of our formulation as a cancer vaccine because it demonstrated high immunogenicity. Having satisfied ourselves with the high immunogenicity of CpG-NP-Tag, in this paper we proved its ability to induce mucosal antitumor immunity that can prevent lung metastases.

Reference:  Kokate RA, Chaudhary P, Sun X, Thamake SI, Maji S, Chib R, Vishwanatha JK, Jones HP. Rationalizing the use of functionalized poly-lactic-co-glycolic acid nanoparticles for dendritic cell-based targeted anticancer therapy. Nanomedicine (Lond). 2016;11(5):479-94. doi: 10.2217/nnm.15.213. Epub 2016 Feb 19. PMID: 26892440; PMCID: PMC55639430

Comment 3: The morphology and dispersity of nano-vaccine should be detected by SEM.

Response: Thank you for the comment. Again, we have previously reported this in published manuscript where we proved the immunogenicity of CpG-NP-Tag as referenced in point 2. We used scanning electron micrograph to show the morphology and dispersity of CpG-NP-Tag.

Comment 4: The CpG loading efficiency of nano-vaccine should be characterized using DNA gel electrophoresis.

Thank you for the comment. Again, we have previously reported this in published manuscript when we proved the immunogenicity of CpG-NP-Tag as referenced in point 2. We used FITC conjugate CpG-NP-Tag to show CpG binding efficiency to NP-Tag.

Comment 5: The resolution of schematics and figures should be improved (e.g., figure 4A and 6A).

Response: Thank you for pointing this out. This has been improved in our revised manuscript.

Comment 6: The manuscript should be carefully checked to avoid the spelling, expression and grammar mistakes, and proofread by a native professor. For instance:

(1) In Figure 1, “4TI”?

(2) In line 589, “IFN-” in the discussion.

Response: Thank you again for pointing this out. These have been duly checked and corrected

Comment 7: For a better presentation, some figures and literal representation needed to be revised.

(1) The fonts of texts should be consistent (e.g., Figure 3a and 3b);

(2) There should be space between word and unit (e.g., line 170, 11000g and 220, 2mM);

(3) In Figure 2C, “AY” should be written in lower case.

Response: Thank for pointing these out. These have been duly checked and corrected.